# Direct visualization of hydrogen absorption dynamics in individual palladium nanoparticles

Tarun C. Narayan[1,*], Fariah Hayee[2,*], Andrea Baldi[1,3,*], Ai Leen Koh[4], Robert Sinclair[1] & Jennifer A. Dionne[1,5]

Many energy storage materials undergo large volume changes during charging and discharging. The resulting stresses often lead to defect formation in the bulk, but less so in nanosized systems. Here, we capture in real time the mechanism of one such transformation—the hydrogenation of single-crystalline palladium nanocubes from 15 to 80 nm—to better understand the reason for this durability. First, using environmental scanning transmission electron microscopy, we monitor the hydrogen absorption process in real time with 3 nm resolution. Then, using dark-field imaging, we structurally examine the reaction intermediates with 1 nm resolution. The reaction proceeds through nucleation and growth of the new phase in corners of the nanocubes. As the hydrogenated phase propagates across the particles, portions of the lattice misorient by 1.5%, diminishing crystal quality. Once transformed, all the particles explored return to a pristine state. The nanoparticles' ability to remove crystallographic imperfections renders them more durable than their bulk counterparts.

[1] Department of Materials Science and Engineering, Stanford University, 496 Lomita Mall, Stanford, California 94305, USA. [2] Department of Electrical Engineering, Stanford University, 496 Lomita Mall, Stanford, California 94305, USA. [3] DIFFER—Dutch Institute for Fundamental Energy Research, De Zaale 20, 5612 AJ Eindhoven, The Netherlands. [4] Stanford Nano Shared Facilities, Stanford University, Stanford, California 94305, USA. [5] Stanford Institute for Materials and Energy Sciences, SLAC National Accelerator Laboratory, Menlo Park, California 94025, USA. * These authors contributed equally to this work. Correspondence and requests for materials should be addressed to J.A.D. (email: jdionne@stanford.edu).

The development of improved energy storage technologies is crucial for the advancement of a number of industries including large-scale alternative energy, clean transport and portable electronics[1]. Two promising strategies—electrical energy storage in batteries and chemical storage of hydrogen in metals—often rely on solute-induced phase transformations[2–4]. These transformations are generally accompanied by large structural changes from incorporation of the solute atom[5,6]. In bulk samples, the large stresses resulting from volume changes cause the formation of several misfit dislocations and eventual fracture, which reduce the cyclability of the system[7–12]. To address these problems, there has been a push towards nanoscale systems, as they have proven to have faster transformation kinetics and are more robust upon repeated charge/discharge cycles[13,14].

Recent work suggests that conducting solute introduction and removal at high rates suppresses phase separation and thus causes the reaction to proceed through a continuous solid solution[15]. The resulting lack of phase coexistence and, accordingly, the lack of interfacial strains or defects, helps to explain the increased durability. At lower rates, however, many systems undergo phase separation and thus sustain large stresses[10,16–21]. How nanoparticles exhibiting solute-induced phase transformations suffer such high stresses but remain durable remains unclear; notably, in typical systems, solute uptake induces significant volume changes on the order of 7–10% (refs 5,6,22).

Palladium hydride serves as an excellent model to understand solute-induced phase transitions. This system is characterized by one of the oldest and most well-studied solute-driven phase transitions, with physics that closely parallel those of Li intercalation and deintercalation compounds, such as $LiNiMnO_4$ (refs 23–25). Moreover, the palladium hydrogen system shows fairly fast kinetics at readily attainable temperatures and pressures, allowing more accessible probing of the phase transformation thermodynamics[26]. $PdH_x$ generally exists in two face-centered cubic phases: a hydrogen-poor $\alpha$ phase existing at lower $H_2$ pressures and a hydrogen-rich $\beta$ phase existing at higher $H_2$ pressures.

The phase transformation behaviour of $PdH_x$ is well known in the bulk, but the changes at the level of individual nanoparticles are only now starting to be addressed, thanks to the development of several single-particle techniques, including in situ transmission electron microscopy (TEM)[23,27,28], plasmonic nanospectroscopy[29,30] and coherent X-ray diffractive imaging[21]. These studies have demonstrated that single crystalline particles do not exhibit phase coexistence at equilibrium[23,29], in contrast to multiply twinned particles[27]. Recent X-ray diffraction experiments have captured intermediates during the hydrogenation reaction, revealing the arrangement of the $\alpha$ and $\beta$ phases and their corresponding strain profiles at one step during the hydrogenation reaction[21]. However, none of these studies reveals the nature by which the $\alpha$ phase transforms to the $\beta$ phase in real time. Here we conduct high-resolution dynamic studies of the $\alpha$ to $\beta$ transformation on the subparticle level. Our results not only give structural insights into the reaction intermediates but help explain the high durability of such nanoparticles in energy storage devices.

An environmental TEM serves as an effective tool to study the hydrogenation of palladium in situ[31,32]. The ability to flow in hydrogen gas at pressures up to 600 Pa with a variable temperature stage allows us to study both structural and spectroscopic properties as a reaction occurs. For example, in palladium, the lattice constant increases by 3.7% and the bulk plasmon resonance shifts by 2 eV upon transformation from the $\alpha$ to the $\beta$ phase[6,23,27,33,34]. Techniques such as selected area electron diffraction (SAED), dark field (DF) imaging, and scanning TEM (STEM) allow insight into the particle structure and crystallography. Electron energy loss spectroscopy (EELS) quantifies the energy lost by the electron beam as it excites a variety of processes, characterizing electronic changes in Pd[35]. Combined, these techniques allow us to image particles with sub-nanometre resolution and allow for thorough structure–function correlation.

Here, we first use a combination of STEM and EELS to image the hydrogen absorption process in single crystalline Pd cubes in real time. We find that the reaction proceeds through a nucleation-and-growth pathway where the $\beta$-phase nucleates in one or more corners of the cube before establishing a (100) phase front. We then examine nanocubes using DF imaging and SAED after freezing the reaction while it is in progress to examine the various reaction intermediates in greater depth (see Supplementary Methods and Supplementary Discussion for details). This analysis suggests the development of a lattice misorientation, which disappears upon completion of the transformation. SAED patterns of representative particles that have been loaded and unloaded twice show that the diffraction spots sharpen upon loading, further underscoring that the completion of the solute absorption process can reverse the crystal quality degradation induced during the $\alpha$ to $\beta$ transformation.

## Results

**Real-time monitoring of the phase transformation.** We start by preparing single crystalline palladium nanocubes using previously published procedures[36,37]. By using the smaller nanocubes as seeds, we prepare cubes ranging from about 15 to 80 nm in size. We deposit the particles on an amorphous $SiO_2$ grid for analysis in the electron microscope and clean them of organic contaminants using a 25% oxygen in argon plasma. A similar cleaning procedure has been shown to leave the surface ligand-free[29]. The $SiO_2$ substrate does not contaminate in the presence of hydrogen and offers a featureless background for EELS in the relevant energy range[38], making it ideal for in situ TEM experiments. The resulting sample is shown in Supplementary Fig. 1.

After introducing the sample into the microscope, we examine it with STEM-EELS in a $H_2$ atmosphere to understand the real-time progression of the hydrogenation reaction. At the voltage and camera length used in this experiment, STEM images show notable diffraction contrast, as elaborated in the Supplementary Methods and Supplementary Fig. 2. Depending on the precise orientation of the particle, either the $\alpha$ or $\beta$ phase can appear brighter. The difference likely stems froms the difference in the excitation error for the spots captured by the annular dark-field STEM detector. By temporarily positioning the beam at different locations in the particle, we collect the EEL spectra and thus assign phases to the regions of differing contrast. An example of this procedure is shown in Supplementary Fig. 3.

STEM time series from three representative particles with edge lengths of 20, 36 and 43 nm are shown in Fig. 1. Regardless of particle size, the reaction begins with the formation of a $\beta$-phase nucleus in one or more corners of the particle. While the orientation of the interface between the $\beta$ nuclei and the $\alpha$-phase matrix cannot be deduced from the STEM images, the diagonal nature of the phase boundary is consistent with a (111)-type interface. This interface has also been shown in a prior study that demonstrates a coherent (111)-oriented phase boundary between the $\alpha$ and $\beta$ phases during early stages of the transformation in an $\sim100$ nm palladium cube[21]. In these cubic particles, the observed morphology of phase nucleation and growth does not follow the spherical shell mechanism previously suggested for palladium nanoparticles[23], but rather resembles the spherical cap model proposed for olivine-based cathodes in lithium-ion batteries[39]. Compared with the spherical shell model, which predicts

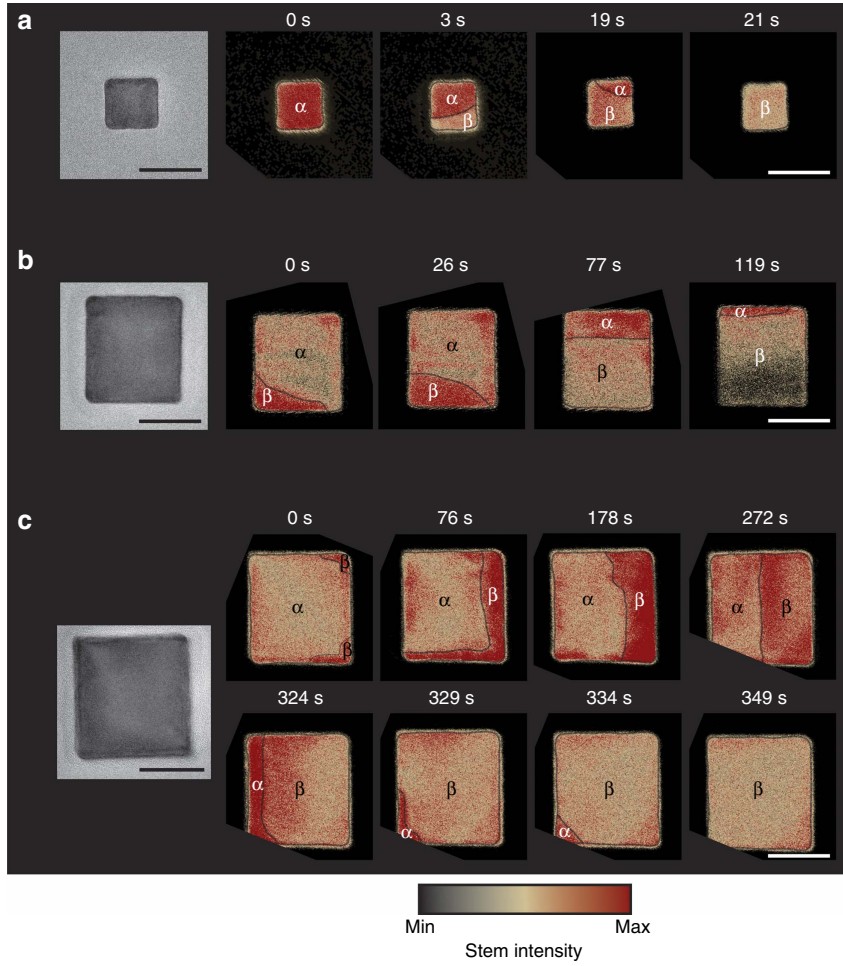

**Figure 1 | Snapshots of scanning transmission electron microscopy movies of the phase transformation.** Still frames of the phase transformation as followed with scanning transmission electron microscopy (STEM) for three different particles of sizes (**a**) 20 nm, (**b**) 36 nm and (**c**) 43 nm accompanied by their respective bright field transmission electron microscope images. The correspondence between each region and its respective phase was verified using electron energy loss spectroscopy. The dotted lines represent the approximate locations of the phase boundaries. In some images, the cube drifts out of the field of view briefly, thus resulting in the image being cut off. The scale bar is 25 nm in each image. The particles shown in **a,b** show the presence of a single nucleus of the β phase, whereas the particle in **c** shows the existence of two nuclei.

coherent phase transitions for particles smaller than 35 nm (ref. 23), the spherical cap model leads to a lower elastic penalty for the coherent existence of an interface between the α and β phases. It is therefore reasonable to assume that cubic nanoparticles larger than 35 nm will still maintain coherency during hydrogen loading and unloading.

The preferential nucleation at the corners of the cubes could have thermodynamic or kinetic origins. In a previous study on the hydrogenation of palladium nanocrystals, a phase field model suggested that the corners of a cuboidal particle are under tensile strain[21]. Such strain lowers the enthalpy of the α- to β-phase transition and promotes preferential hydrogen uptake in the corners. Prior work has also shown that hydrogen absorption into 111-terminated octahedra occurs more quickly than into 100-terminated cubes[40]. It is therefore likely that hydrogen absorption will be faster through the '111-like' surfaces at the rounded corners of the cubes, which would again facilitate hydrogen uptake through the particle corners.

In the two smaller cubes shown in Fig. 1a,b, only one β-phase nucleus appears to form. The nucleus in the 36 nm cube shown in Fig. 1b in particular seems to have a (111) type interface at the corner, but spills out across the bottom edge of the cube to nearly establish a (100) phase front across the particle. In the first 26 s, it

appears that the β-phase nucleus does not grow horizontally, but grows diagonally, suggesting that the phase front moves faster along ⟨111⟩ or ⟨110⟩ than it does along ⟨100⟩. Once the phase front is established, as in the image taken after 77 s, it is very stable and does not suffer from reorientations. An interface oriented along (100) is consistent with a coherent process, as a (100) phase boundary minimizes the elastic energy penalty required to establish a coherent interface in an infinite medium (as elaborated in the Supplementary Discussion and Supplementary Fig. 4)[41]. This prediction corroborates earlier TEM work on β phase coherent precipitates in a palladium foil[42], suggesting that the α to β transition proceeds coherently. Once the phase front nears the edge of the particle, the α phase is confined to the corners of the nanocube and eventually squeezed out.

The smaller cube in Fig. 1a exhibits many similar tendencies. Since the particle is smaller, the β phase spreads across one face of the cube more quickly and establishes a {100} phase front before the {111} phase front has finished growing, as suggested by the curved interface. The final steps closely resemble those of the 36 nm cube.

In the larger particle shown in Fig. 1c, the β-phase nuclei in the opposite corners of the cube connect to form the (100) interface.

The phase front initially reorients itself continuously but eventually stabilizes in one of the $\langle 100 \rangle$ directions. The β-phase front then begins to propagate across the particle slowly. For example, from 0 to 76 s, approximately 20% of the particle has converted from the α to the β phase. In the next 200 s, approximately 50% of the particle has transformed, meaning that 30% of the particle has been hydrogenated in that time period; the transformation rate during this time has reduced by more than a factor of two. As such, we see that growth is faster at earlier stages, so the phase transition is unlikely to be surface limited as the (100) phase front moves across the particle. In addition, we note that the diffusion constants of hydrogen in the α and β phases at about 250 K are $7 \times 10^{-8} \, cm^2 \, s^{-1}$ and $2 \times 10^{-8} \, cm^2 \, s^{-1}$, respectively, meaning that hydrogen could travel 100 nm in under 1 ms (refs 26,43). The rate of the α-to-β phase transformation is thus not limited by hydrogen diffusion or surface reactions, but by the slow motion of the phase boundary.

After a slow progression through the bulk of the particle, the later stages of the phase transformation occur very quickly. Upon nearing the edge of the particle, the α phase is restricted to a corner of the particle, as seen in the 329 and 334 s snapshots in Fig. 1c. The morphologies of the two phases in this stage resemble those seen during the initial stages of the reaction, implying that the beginning and end of the reaction occur by a similar mechanism.

An interesting feature of some of the STEM movies is that, in some cases, the contrast inverts between the α and β phases; the correspondence between regions of different contrast and their EEL signatures has been observed as in Supplementary Fig. 3. This effect can be seen between the 46 and 77 s time steps of Fig. 1b and the 272 and 324 s time steps of Fig. 1c. In both cases, the contrast inversion occurs as the β phase propagates across the sample. Since the features in our STEM images arise from diffraction contrast, the contrast inversion could arise from a lattice reorientation and/or deformation. This hypothesis is futher supported by high-resolution electron diffraction, as described below.

**Structural analysis of reaction intermediates**. With this general understanding of the dynamic phase transformation, we then investigated the structure of the reaction intermediates to gain more detailed insight into the mechanism. To do so, we used a combination of displaced-aperture dark field (DADF) imaging and SAED. We first rapidly increased the hydrogen partial pressure to 500 Pa at $-35\,°C$, which is slightly above the pressure required for the α to β transformation. After about 2 min, the nanoparticles were rapidly cooled to 100 K, while slowly purging the $H_2$ gas from the sample compartment. At 100 K, the palladium surface is catalytically inactive towards $H_2$ splitting and recombination, and we therefore effectively trap the hydrogen atoms inside the particles[44].

Since the diffraction spots shift inward by approximately 3.5% upon transitioning to the β phase, spots arising from the two phases are easily resolvable. Introducing an objective aperture allows for the selection of electrons diffracting from a single phase; these electrons can then be used to generate a DADF image, as seen in Fig. 2a. Overlaid dark-field images collected from the outside (blue) and inside (red) diffraction spots of 24 different particles are shown in Fig. 2b–g. The particles are arranged in the order of increasing hydrogen content within particles of similar sizes. The dark-field image constructed using both the α- and β-phase spots is shown in Supplementary Fig. 5. We see immediately that the DF images closely parallel those seen in the dynamic STEM measurement, and that cubes of different sizes transform by a similar mechanism. Nucleation of the β

phase appears to occur at the corners of the cube as seen in columns i and ii. The cubes we observed largely had two or more nuclei, especially in the case of larger cubes, which suggests that nucleation and movement of the phase boundary occur with comparable rates. We also note that the striping seen in the β-phase nuclei, most prominently in the second column of Fig. 2c, are consistent with intensity fringes due to a varying thickness of these crystallographic domains[45]. A varying thickness is consistent with the spherical cap model, where the β-phase cap thickness progressively decreases away from the cube corners along $\langle 110 \rangle$. The nucleation step is followed by growth and coalescence of nuclei into a (100) phase front seen in columns ii and iii, which then propagates across the particle and, in some cases, leaves a small phase region in the corner of the particle as seen in the fourth column of Fig. 2c,d.

Figure 3 displays representative diffraction patterns corresponding to the dark field images shown in Fig. 2d. The first column of Fig. 3b shows the diffraction pattern for a cube with a (111)-type interface. A zoomed-in image of the 020 spot is shown in the first column of Fig. 3c. We can see that there are only two spots—one corresponding to the β phase and one corresponding to the phase—that essentially lie on the same radial line.

The dark-field image in the second column of Fig. 3a corresponds to the growth of the β-phase nuclei to the point in which they begin to form the (100) phase front. The two growing β-phase nuclei are similar in appearance to the β-phase nucleus in Fig. 1b at 46 s. In this case, the diffraction pattern in the second column of Fig. 3b, and the corresponding zoomed-in spot in Fig. 3c, is similar to that seen in the first panel, suggesting that coherency is likely maintained at this stage. The third column of Fig. 3b shows a representative diffraction pattern of a particle that is roughly 50% transformed into the β phase. Each spot appears to have smeared out into a streak. Closer examination of an individual spot as in the third column of Fig. 3c shows that each spot has split into two α–β pairs. A fit of the two sets of lattices shows that one is rotated by approximately 1.5° with respect to the other. This secondary set of spots implies that a portion of the lattice exists in a slightly different orientation than the rest of the particle. Our findings are consistent with an earlier report of the hydrogenation of palladium films, in which the authors find that a majority of the crystallites are rotated 1.5–3° with respect to their original orientation[46]. The rotation in the diffraction pattern corresponds to the rotation in the plane of the substrate. A slight additional out-of-plane component, which is difficult to detect using this pattern, could alter the diffraction condition and thus contribute to the change in contrast that is observed in the STEM images. There is not enough evidence to interpret the nature of this rotation to distinguish between phenomena such as dislocations or strained rotations. Recent calculations and experiments in highly mechanically stressed nanoscale systems have shown that the barrier towards creation of partial dislocations decreases with decreasing particle size, although the spherical cap morphology seen here suggests that there may not be enough of a driving force to nucleate a dislocation[39,47,48].

As the α phase becomes isolated in a corner of the particle as in the fourth column of Fig. 3a, the diffraction spot in Fig. 3c now resembles that taken at the early stage of the transformation; the only difference is that now the β-phase spots are more intense than the α-phase spots. There are no longer two sets of lattices in the particle that show different orientations. This absence, along with the close resemblance of the frozen-in intermediates with the real-time images, suggests that the particles remove crystallographic imperfections during the loading process, as the misoriented part of the nanoparticle either realigns with the rest of the particle or the phase front pushes the rotated region out of the particle. The diffraction patterns of other particles with

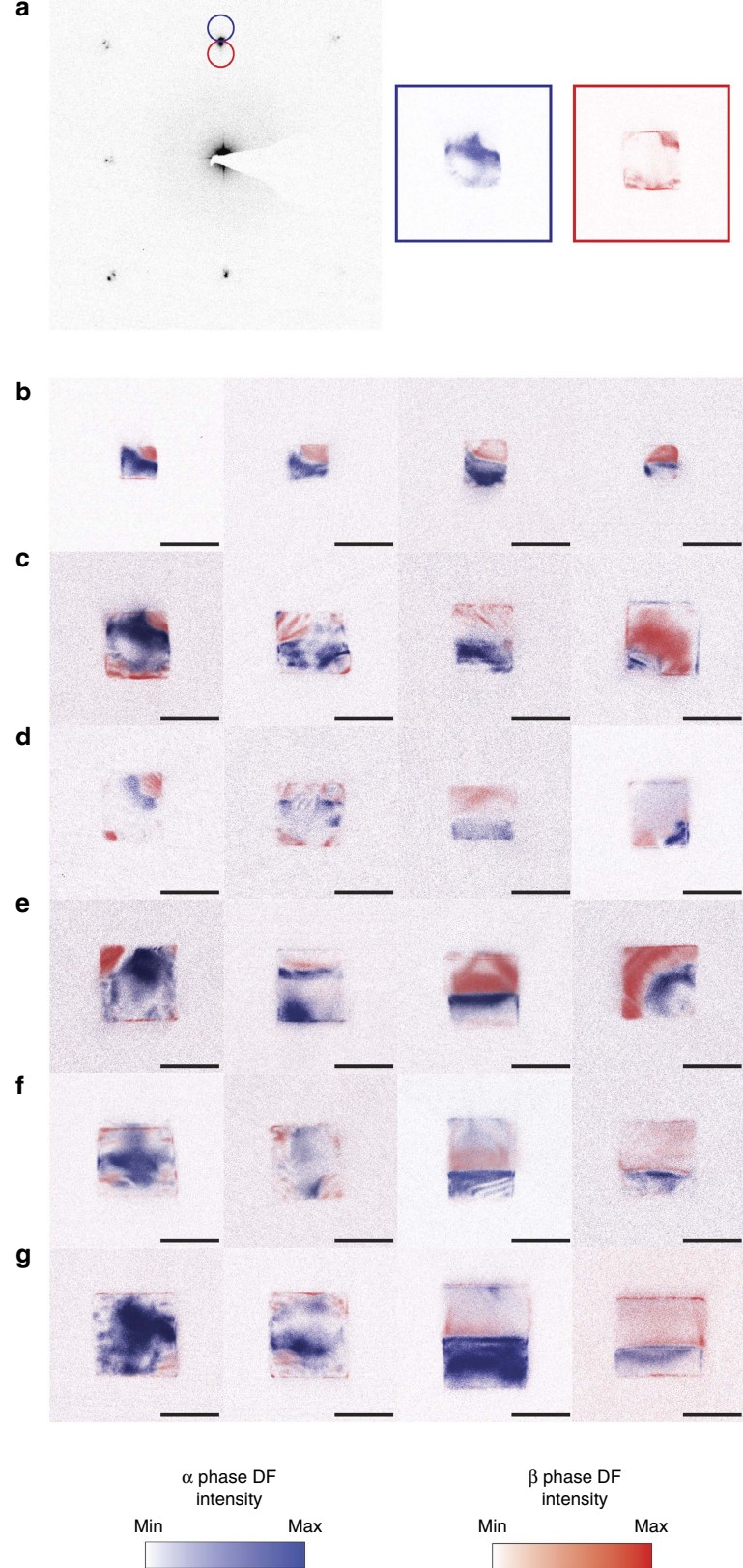

**Figure 2 | Dark field images of reaction intermediates.** (**a**) Representative electron diffraction pattern. The red and blue circles correspond to the positions of the objective apertures that give rise to the images shown to the right. (**b–g**) Overlaid dark field images of 24 different particles grouped together by size range obtained from the outside spot (blue) and the inside spot (red). Within each row, the particles are approximately arranged by increasing transformed fraction to attempt to recreate the phase transformation. The scale bar is 50 nm in each image.

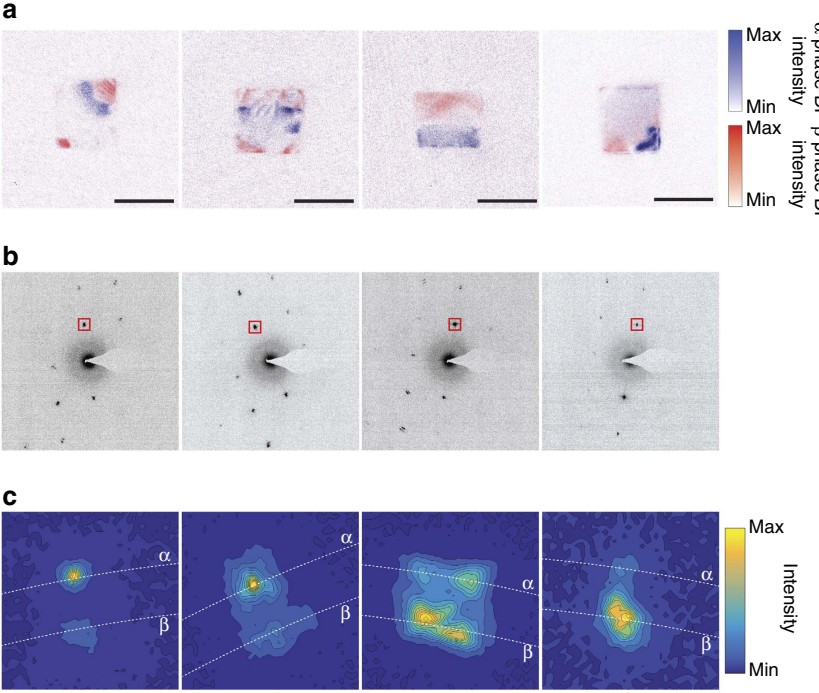

**Figure 3 | Diffraction patterns of selected reaction intermediates. (a)** Overlaid dark field images shown in Fig. 2d. The scale bar is 50 nm in each image. **(b)** Diffraction patterns corresponding to the dark field images are shown. **(c)** The diffraction patterns shown in **b** zoomed in on the region highlighted with the red square. The white dotted lines correspond to the arc delineating constant lattice parameter.

similar morphologies lead to the same conclusions. Crystallographic 'healing' in nanoparticulate phase transformations has also been observed in the $LiNi_{0.5}Co_{0.5}O_2$ system[10]. In this case, X-ray imaging shows that edge dislocations formed by the coexistence of phases appear to migrate to the surface of a particle during lithiation. Unlike the bulk, in which the defect is trapped inside, nanoparticles have a nearby surface to which defects can migrate and reduce energy penalties required to annihilate dislocations.

**Equilibrium analysis of hydriding and dehydriding.** To check that the continual probing of particles with the electron beam does not severely alter the reaction process, we investigate nanocubes after they have reached equilibrium. Pressure-composition isotherms are collected using a series of SAED patterns at different hydrogen pressures for particles with edge lengths of 19, 33, 48 and 74 nm, as shown in Fig. 4a. At each pressure, the system is allowed to settle for 30 min, consistent with the equilibration times shown in Supplementary Fig. 6. These isotherms both confirm that the particle remains in a single phase at equilibrium and remains single-crystalline through both the α to β and β to α phase transitions. We see no evidence of a secondary rotated lattice. The retention of crystallinity is consistent with our earlier findings[23].

The isotherms, however, do not capture crystal quality during the course of the reaction. To better understand the impact of the phase transition on this parameter, we cycle the particles with hydrogen and monitor the average change in the peak width of the observable spots as referenced to the diffraction pattern of the pristine particle. A representative data set is shown in Fig. 4b for a 31 nm particle (similar plots for 19 and 47 nm particles are shown in Supplementary Fig. 7). The spots become noticeably sharper after the α to β phase transformation; this behaviour is seen consistently upon cycling. Since the reciprocal of the spot width is approximately proportional to the crystallite size, crystal quality

increases during the loading process. The cycling data thus confirm that the hydrogen absorption process serves as a mechanism to remove crystallographic imperfections in the particle.

Our results corroborate recent work proposing a coherent loading process but an incoherent unloading process[49]. The morphologies we observe are consistent with coherency at the beginning and end of the reaction; furthermore, the spot size at the end of the transformation indicates that the reaction proceeds without formation of many dislocations. We see such results for all particle sizes observed, suggesting that bulk-like incoherent transitions are not present at least up through sizes of ~80 nm for the loading process. Upon desorption, however, the broadening of the diffraction spots shows the process occurred with significant defect formation, potentially indicative of an incoherent mechanism.

Using a combination of SAED, DF imaging, STEM and EELS in an environmental TEM, we are able to follow the hydrogenation of palladium *in situ* and discern the mechanism of the process. Regardless of size, the particles exhibit similar dynamic behaviour. The β phase initially nucleates at one or more corners of the cube. The nucleus then grows across the cube after establishing a (100) phase front. During this process, a portion of the transforming particle reorients, thus giving rise to four or more crystalline domains in the initially single-crystalline particle. After moving across the particle, the β phase segregates the α phase to a corner of the cube, at which point the particle no longer contains reoriented regions. Cycling experiments consisting of two loading and unloading processes confirm that the particle improves its crystal quality during the α to β phase transformation. The ability of a nanoparticle to remove prior imperfections allows it to be an effective medium for energy storage, as it is able to maintain a high degree of crystallinity during the cycling process. Our results demonstrate the utility of nanoscaling for solute-based phase transformations—the small size allows for defects to be pushed out of the particle, which is not possible in the bulk.

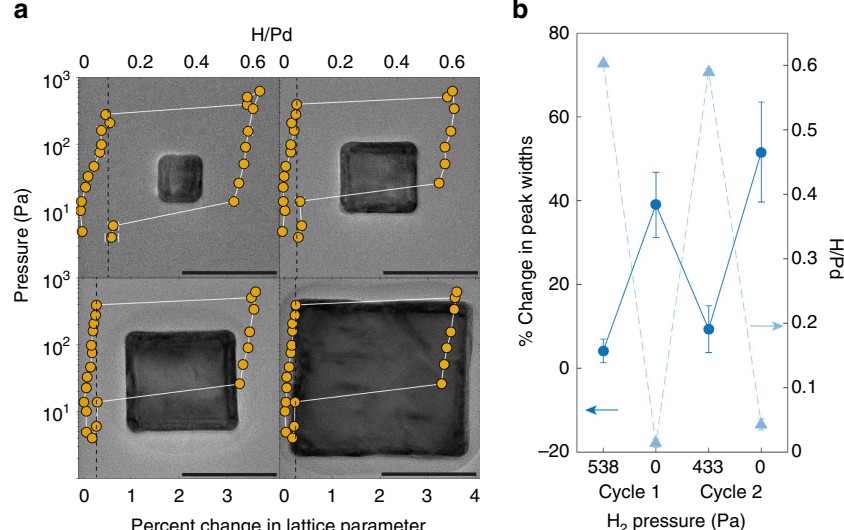

**Figure 4 | Diffraction-based pressure-composition isotherms.** (**a**) Representative pressure-composition isotherms collected using electron diffraction for four different particle sizes. The transmission electron microscopy images in the background of each isotherm were taken after two cycles of hydrogenation and dehydrogenation. The separations between pairs of spots are represented as percentage changes from the separation in the reference pattern of the unloaded state. The error bar corresponds to the standard error of the distribution of changes in separation. The scale bar is 40 nm in each image. (**b**) Change in the width of diffraction spots upon cycling. The width of each spot is referenced to the original spot in the room temperature diffraction pattern. The standard deviation of this distribution of width changes corresponds to the error bar. The circular points correspond to the percentage change in spot width and are thus referenced to the left axis. The triangular points correspond to the hydrogen loading and are thus referenced to the right axis. Note that the average peak width decreases upon loading and increases upon unloading, suggesting that the hydrogen absorption process increases crystallinity.

**Data availability**. All relevant data are available from the authors.

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

## Acknowledgements

We gratefully acknowledge scientific feedback and discussions with all Dionne group members and Professor Ronald Griessen. J.A.D. acknowledges support from a PECASE Award administered by the Air Force Office of Scientific Research (FA9550-15-1-0006) and a National Science Foundation CAREER Award (DMR-1151231). This work was supported in part by a SLAC National Accelerator Laboratory LDRD award in concert with the Department of Energy, Office of Basic Energy Sciences, Division of Materials Sciences and Engineering, under contract DE-AC02-76SF00515. Work was also supported by the research program 'Fellowships for Young Energy Scientists' (YES!) of the Foundation for Fundamental Research on Matter (FOM), which is financially supported by the Netherlands Organisation for Scientific Research (NWO) and by an award from the Department of Energy (DOE) Office of Science Graduate Fellowship Program administered by the Oak Ridge Institute for Science and Education for the DOE. ORISE is managed by Oak Ridge Associated Universities (ORAU) under DOE contract number DE-AC05-06OR23100. All the opinions expressed in this paper are the author's and do not necessarily reflect the policies and views of DOE, ORAU or ORISE. Part of this work was performed at the Stanford Nano Shared Facilities (SNSF).

## Author contributions

All authors contributed to the design of the experiment. T.C.N., F.H., A.B. and A.L.K. carried out the experiment. T.C.N. and F.H. wrote the first draft of the manuscript and all the authors assisted in the writing process and data analysis.

## Additional information

**Competing financial interests:** The authors declare no competing financial interests.

**Publisher's note**: 

