## [Peer Review File · Nature Communications]

Reviewers' comments:

Reviewer #1 (Remarks to the Author):

In general, this is an excellent paper that is well written, structured, and features beautiful images. The resolution and image quality is outstanding. It is also great to see the use of electron diffraction in the analysis. The paper is definitely novel and important.

I have one major concern:

The creation and healing of defects. The authors observe a lattice rotation between the alpha and beta phases of approximately 1.5 degrees, which is consistent with previous reports. They then interpret this rotation to mean that the particle has formed a defect. I think a simpler explanation is that the lattice maintains coherency and achieves this rotation through strain rather than the introduction of some sort of dislocations. The reason that I am more inclined to believe in a strained rotation is that defect nucleation is typically thought of as an irreversible process. The authors do not observe the defect in the fully hydrides phase (consistent with ref. 21). To nucleate a defect requires significant energy, which may be available, but then to have annihilation of the defect would again require energy and it is not clear how this happens. The fact that nanoparticles can do this rotation and not nucleate a defect is probably due to their relatively high surface to volume ratio.

Finally, it seems unlikely that there is enough strain energy to have the initial defect nucleation. In recent papers (Baldi et al 2014, Griessen et al 2015) the critical size is estimated at 30-45 nm using a core-shell model. However, the images shown in Fig. 1 show that the process is not core-shell (and indeed, this seems logical to me as core-shell will not minimize interfacial area). A spherical cap type phase morphology will have much less elastic energy available for dislocation nucleation (one estimate is a factor of 10/3 less, see Fig. 6 of the following reference: Meethong, N.; Huang, H.-Y. S.; Speakman, S. A.; Carter, W. C.; Chiang, Y.-M. Strain Accommodation during Phase Transformations in Olivine-Based Cathodes as a Materials Selection Criterion for High-Power Rechargeable Batteries. *Adv. Funct. Mater.* 2007, 17 (7), 1115-1123.). Besides, the particle size in Fig. 1a is below the critical size assuming core-shell.

Last point regarding the defects. Fig. 3c seems to show intermediate diffraction intensity between the alpha and beta phases, which implies regions that have a lattice constant between alpha and beta. This is consistent with a smoothly varying lattice and no defects.

A couple of minor comments are given below:

Can the authors comment on how many cubes showed nucleation at one corner vs. nucleation at two corners? Was there any trend with the size for this process? In other words, did larger cubes show preference for a single corner vs. multiple corners or did it appear to be somewhat random (potentially due to slight differences in geometry of the cubes)?

For the "freezing" process, I worry that the thermodynamics may be more complicated. For example, a solid solution at high temperature can decompose into a two phase solution at low temperature. Can the authors comment a bit more on this?

The moving of the cube in Fig. 1c 272s is a little troubling. Can the authors be sure that the cube does not shift (which may change the apparent locations of the alpha and beta phase? Or, did they use other analysis such as image registration to correct for this?

In Fig. 2b-g, some of the images have no alpha or beta phase intensity (for example, Fig. 2d part i in the center of the crystal). Is this because that portion of the crystal is rotated out of the diffraction condition? How was the alignment done for these diffraction measurements? Is there evidence of the coherent interface in these maps, i.e. diffraction intensity at intermediate lattice constants between the alpha and beta phase lattice constants? Actually I think this question is answered by Fig. 3c but want to be sure.

Also, are all of the diffraction spots used to construct the phase morphology map or only the brightest one, which I assume is 440.

Fig. 4b, the x axis is a little confusing. Why does it increment as 538 0 433 0 ?

The observation of phase morphology (spherical cap type) in a nanoparticle during a reaction is really impressive. I think the authors should highlight this aspect of the work a bit more.

Supplementary material corresponding author has a email address typo.

Fig. S1 shows that there are rods in the synthesis batch. Did the authors look at the phase morphology in these rods?

Small changes that I would like to see implemented:

- In the caption of Fig. 1, please discuss the different panels (a, b, etc). Are all scalebars the same?
- In the introduction line 39-41 a comparison between Pd-H and LFP and LTO is made. A more comparable battery electrode is actually LiNiMnO_4 , which has both solid solution and two-phase behavior (like Pd-H). LFP and LTO are typically thought of as exclusively two phase systems.

Clarification questions:

- for the Fig. 2 results. I assume the electron beam is rastered across the particle and the diffraction taken at each point in order to make the images in Fig. 2b-g?

Reviewer #2 (Remarks to the Author):

The manuscript 'Direct visualization of hydrogen absorption dynamics in individual nanoparticles' the authors report on a fascinating experimental in-situ work using ETEM and EELS. The authors study the hydride formation of individual single-crystalline Pd-nanocubes ranging from 15 nm to 80 nm in edge length. According to these results the hydride first forms in the cube corners and then the front propagates mainly parallel to the surface planes, e.g. in $\langle 100 \rangle$ -direction. This can be seen for all nanocube sizes, even for the smallest ones. The interface between the two phases inside the cube contains defects. This is supported by the finding of two well-separated hydride peaks in the selected diffraction pattern which are rotated by about 1.5° with respect to the other. The defects are gone after phase transformation. A similar behavior was found for all nanocube sizes. This is the first time of direct visualization of hydride fronts inside individual nanocubes of sizes down to 15 nm. While hydride nucleation in corners and two-phase coexistence in Pd nanocubes of about 80 nm side length and larger was reported via CXDI method by Ulvestad et al. recently, the here shown ETEM results step strongly Forward. They are on much smaller nanocubes sizes and get new insights. They hint on defects accompanying the propagating front, surprisingly even for the small nanocubes. These defects affect the energy required to drive the phase transformation and are, thus, important for application. The results verify that the sub-surface layer which covers the whole nanocube (maybe not the bottom) was discussed to stimulate the hydride growth, has minor impact on the transformation behavior, as a planar front propagates in the nanocube. The results are very exciting and also, as I

assume, difficult to achieve from the experimental part. The formation process is interesting and of general interest, not only for hydride formation in Pd-nanocubes – but also for other nanomaterials that sorb atoms on interstitial sites or by intercalation. For all these reasons I strongly support publication of this manuscript in Nature Materials.

Minor comments:

Abstract: The present version of the abstract highlights too much on the aspect that defects 'self heal' after transformation – which is from my perspective not surprising, as the defects escort the front in the region of heaviest deformation. Once the front has passed the nanocube, the defects (I assume the authors mean a dislocation?) should also have left the sample. On the other hand it is known that defects are not stable in nanosized samples and have extremely fast kinetics. It is even difficult to visualize them in tensile test experiments on free samples – just reacted dislocations can be seen inside the samples (see for ex. B. Roos, B. Kapelle, G. Richter and C. A. Volkert, Appl. Phys. Lett. 105, 201908 (2014)).

Lines 69-82: This part informs about the results in advance. It is too lengthy and the reader is expecting experimental data first. This part further leads to information duplication. The authors should avoid this part.

The impact of strain energy on the phase transformation behavior was recently modelled by Wagner et al. Int. J. Hydr. Energy 41, 2727(2016), on Pd-H thin films. The contribution of the strain energy to the isotherm shifts the chemical potential and affects the hydride formation tremendously – this explains why nanocube corners are hydride nucleation centers and why it is important to keep the system mainly stress-free for storage applications.

Reviewer #3 (Remarks to the Author):

The manuscript describes a detailed in-situ and ex-situ TEM/STEM characterization of the hydrogenation of palladium nanoparticles. The work is a fantastic illustration of what can be accomplished by in-situ TEM if it is performed under controlled illumination conditions. The results are extremely well presented, with the summary in the main text and the details of how the experiments were performed in the supplementary (every question I had about the experiments in the main text I found answered in the supplementary). The paper outline the experimental conditions and what assumptions go into the analysis in full detail that will allow anyone to reproduce the results. The analysis and the conclusions are all that one could want and expect.

Although I feel the paper should definitely be published, there is one area where I thought the authors could include a better discussion in the manuscript:

The paper starts with a description of the need to go to nanoparticles because of the ability to remove defects. However, in the results of figure 4 and supplementary data, there doesn't seem to be much of an effect on cycling in the particle size ranges the authors studied - they all appear to be small enough. What would be the maximum size where the authors feel there would be an effect? Would they be able to see a rate change or structure change specifically with their in-situ/ex-situ methods or would there have to be interpretation based on other results (in the manuscript, mechanisms proposed from other methods are used to corroborate the observations).

Reviewers' comments:

Reviewer #1 (Remarks to the Author):

In general, this is an excellent paper that is well written, structured, and features beautiful images. The resolution and image quality is outstanding. It is also great to see the use of electron diffraction in the analysis. The paper is definitely novel and important.

We thank the reviewer for the positive assessment of our work and agree that the combination of techniques inside the TEM yields important insights into material behavior.

I have one major concern:

The creation and healing of defects. The authors observe a lattice rotation between the alpha and beta phases of approximately 1.5 degrees, which is consistent with previous reports. They then interpret this rotation to mean that the particle has formed a defect. I think a simpler explanation is that the lattice maintains coherency and achieves this rotation through strain rather than the introduction of some sort of dislocations. The reason that I am more inclined to believe in a strained rotation is that defect nucleation is typically thought of as an irreversible process. The authors do not observe the defect in the fully hydrides phase (consistent with ref. 21). To nucleate a defect requires significant energy, which may be available, but then to have annihilation of the defect would again require energy and it is not clear how this happens. The fact that nanoparticles can do this rotation and not nucleate a defect is probably due to their relatively high surface to volume ratio.

We agree that there is not enough evidence to suggest the presence of a dislocation. We note, however, that there also is not enough evidence to suggest that the lattice rotation observed arises due to strain. As we do not have sufficient proof to favor either option, and experiments to do so are sufficiently lengthy and challenging to be outside the scope of this paper, we have omitted the claim that the loading process produced defects. We have also removed most claims of self-healing behavior, and have only included that description to highlight the improvement in crystal quality upon loading.

We have removed the phrase

“We find that the hydrogen absorption process proceeds through a nucleation and growth pathway that tends to form defects as the hydrogenated phase...”

and replaced it with

“We find that the hydrogen absorption process proceeds through a nucleation and growth pathway in which the new phase forms in one or more corners of the particle...”

We have replaced:

“We find that phase coexistence motifs closely match those observed in real-time, indicating that the frozen-in intermediates can be studied to understand the structural transformations occurring

during hydrogen uptake. Analysis of the SAED patterns of the intermediate states shows that the originally single crystalline particles develop defects once the $\langle 100 \rangle$ phase front forms. At the end of the transformation, however, the defects are no longer present.”

with

“This analysis suggests the development of a lattice misorientation, which disappears upon completion of the transformation.”

We have replaced:

“...further underscoring that the solute absorption process can serve to heal the defects induced by the α to β transformation”

with

“...further underscoring that the solute absorption process can reverse the crystal quality degradation induced by the α to β transformation”

We have removed:

“Since the SAED aperture selects only a single particle in our study, the diffraction pattern indicates that the lattice in one portion of the particle is rotated with respect to that in the rest of the particle. The existence of lattices of different orientations suggests that the initially single-crystalline particle has developed a defect during the course of the phase transformation.”

We have added:

“There is not enough evidence to interpret the nature of this rotation to distinguish between phenomena such as dislocations or strained rotations. Recent calculations and experiments in highly mechanically stressed nanoscale systems have shown that the barrier towards creation of partial dislocations decreases with decreasing particle size, although the spherical cap morphology seen here suggests that there may not be enough of a driving force to nucleate a dislocation”

We have replaced:

“... suggests that the particles self-heal during the loading process...”

with

“... suggests that the particles remove crystallographic imperfections during the loading process...”

We have removed:

“Even though the reaction temperature is -35°C during the loading process, the driving force to remove defects is strong enough to return the particle to a single-crystalline state.”

We have replaced:

“Since the reciprocal of the spot width is approximately proportional to the crystallite size, defects appear to be removed during the loading process. The cycling data thus confirm that the hydrogen absorption process serves as a mechanism to heal away defects formed in the particle”

with

“Since the reciprocal of the spot width is approximately proportional to the crystallite size, crystal quality increases during the loading process. The cycling data thus confirm that the

hydrogen absorption process serves as a mechanism to remove crystallographic imperfections in the particle”

We have replaced:

“Cycling experiments confirm that the particle self-heals during the α to β phase transformation.” with

“Cycling experiments consisting of two loading and unloading processes confirm that the particle improves its crystal quality during the α to β phase transformation.”

Finally, it seems unlikely that there is enough strain energy to have the initial defect nucleation. In recent papers (Baldi et al 2014, Griessen et al 2015) the critical size is estimated at 30-45 nm using a core-shell model. However, the images shown in Fig. 1 show that the process is not core-shell (and indeed, this seems logical to me as core-shell will not minimize interfacial area). A spherical cap type phase morphology will have much less elastic energy available for dislocation nucleation (one estimate is a factor of 10/3 less, see Fig. 6 of the following reference: Meethong, N.; Huang, H.-Y. S.; Speakman, S. A.; Carter, W. C.; Chiang, Y.-M. Strain Accommodation during Phase Transformations in Olivine-Based Cathodes as a Materials Selection Criterion for High-Power Rechargeable Batteries. *Adv. Funct. Mater.* 2007, 17 (7), 1115-1123.). Besides, the particle size in Fig. 1a is below the critical size assuming core-shell.

We thank the referee for pointing us towards the spherical cap strain energy calculations. We agree that the similar behavior of the cubes even up to 80 nm suggests they transform by the same mechanism, in contrast to our earlier estimate for critical size. We have correspondingly updated the text to reflect this information. We do not, however, rule out the possibility of the formation of dislocations or stacking faults. The interfacial stress at the phase boundary is likely on the order of GPa. It has been shown that aluminum, another metal with a high stacking fault energy like palladium, undergoes a large reduction in the defect formation energy upon nanosizing (Carlton, *Philos. Mag. Lett.* 2008). The analysis only considers the effect of two surfaces, so it is possibly that the inclusion of four additional surfaces could further lower the barrier to partial dislocation nucleation. Additionally, the observation of twin planes and stacking faults in single crystalline gold wires under tension suggest that nanoscale objects can plausibly nucleate dislocations at their surface (Roos, *Appl. Phys. Lett.* 2014). Single nanoparticles have also been shown to nucleate dislocation loops under stress (Carlton, *Microsc. Microanal.* 2007).

We have added:

“In these cubic particles, the observed morphology of phase nucleation and growth does not follow the spherical shell mechanism already suggested for palladium nanoparticles, but rather resembles the spherical cap model proposed for olivine-based cathodes in lithium-ion batteries. Compared to the spherical shell model which predicts coherent phase transitions for particles smaller than 35 nm, the spherical cap model leads to a lower elastic penalty for the coherent existence of an interface between the α and β phases. It is therefore reasonable to assume that cubic nanoparticles larger than 35 nm will still maintain coherency during hydrogen loading and unloading.”

Last point regarding the defects. Fig. 3c seems to show intermediate diffraction intensity

between the alpha and beta phases, which implies regions that have a lattice constant between alpha and beta. This is consistent with a smoothly varying lattice and no defects.

We agree that the intermediate intensity between peaks is reasonable evidence for the existence of a coherent interface. However, since the peaks can also be wide due to the small crystallite size, we do not make the strong claim that the diffraction pattern is proof of coherent behavior.

A couple of minor comments are given below:

Can the authors comment on how many cubes showed nucleation at one corner vs. nucleation at two corners? Was there any trend with the size for this process? In other words, did larger cubes show preference for a single corner vs. multiple corners or did it appear to be somewhat random (potentially due to slight differences in geometry of the cubes)?

Most particles showed nucleation at multiple corners. The smallest cubes (less than 30 nm) showed nucleation at a single corner exclusively. Of the remaining 14 cubes examined, only two showed nucleation at a single corner. The rate of nucleation is thus likely slightly lower than the rate of growth, as it seems that only larger particles have time to establish two nuclei before the 100-type phase front is established.

For the "freezing" process, I worry that the thermodynamics may be more complicated. For example, a solid solution at high temperature can decompose into a two phase solution at low temperature. Can the authors comment a bit more on this?

We absolutely agree that a high temperature solid solution could complicate our data analysis. We note, however, that in our dynamic STEM data, we observe two-phase morphologies that resemble those observed in the low-temperature dark-field data. It is of course difficult to discern solid solutions using the STEM imaging technique, so it is possible that the particles, for instance, start in a supersaturated solid solution before decomposing. Our only claim is that, in the stage of the reaction that we observe with STEM, there appears to be a relatively clear boundary, suggesting that the particle is in a two-phase state. Based on this assumption, we propose that the contribution of the solid solution state is unlikely to be a significant contributor (volumetrically) to the reaction mechanism. Furthermore, high-temperature solid solutions can decompose into low-temperature two-phase mixtures if the system crosses the critical temperatures upon cooling. For the nanoparticles we investigate in this paper (15-80 nm) the size-dependent critical temperature for the α to β phase transition in palladium is always greater than 400 K. All of the experiments described in the paper are done at temperatures well below these temperatures and the thermodynamics are therefore not expected to dramatically change upon cooling.

The moving of the cube in Fig. 1c 272s is a little troubling. Can the authors be sure that the cube does not shift (which may change the apparent locations of the alpha and beta phase? Or, did they use other analysis such as image registration to correct for this?

It is unlikely that the particles shift significantly on the substrate. As we monitored the reaction in several particles, we never saw motion of one particle with respect to the other particles on the

substrate. In some cases, the image contained a portion of a second particle (which we cropped out for clarity). The relative position of the two particles with respect to each other did not change.

In Fig. 2b-g, some of the images have no alpha or beta phase intensity (for example, Fig. 2d part i in the center of the crystal). Is this because that portion of the crystal is rotated out of the diffraction condition? How was the alignment done for these diffraction measurements? Is there evidence of the coherent interface in these maps, i.e. diffraction intensity at intermediate lattice constants between the alpha and beta phase lattice constants? Actually I think this question is answered by Fig. 3c but want to be sure.

The particles were not rotated before image acquisition. It is possible that the region of the particle most strongly satisfying the Bragg condition is the interface region in the particles showing minimal intensity at the center, suggesting the strain present at the boundary leads to a slightly different diffraction condition.

There is certainly diffraction intensity between peaks, but we hesitate to claim it as evidence for a coherent process. The intensity could also arise from the relatively wide peaks coming from the nanocrystals and the small molar volume difference between the α and β phases.

Also, are all of the diffraction spots used to construct the phase morphology map or only the brightest one, which I assume is 440.

Only the brightest spot is used to construct the phase map. The diffraction spot used for different particles varies because the particles all have slightly different orientations and we did not reorient them along a particular direction.

Fig. 4b, the x axis is a little confusing. Why does it increment as 538 0 433 0 ?

The axis shows that we are cycling the gas pressure between vacuum and “high” pressure. We have modified the figure to include labels for “cycle 1” and “cycle 2.”

The observation of phase morphology (spherical cap type) in a nanoparticle during a reaction is really impressive. I think the authors should highlight this aspect of the work a bit more.

We agree that we did not focus on this finding enough. We have updated the paper to better highlight this.

We have added:

“In these cubic particles, the observed morphology of phase nucleation and growth does not follow the spherical shell mechanism already suggested for palladium nanoparticles, but rather resembles the spherical cap model proposed for olivine-based cathodes in lithium-ion batteries. Compared to the spherical shell model which predicts coherent phase transitions for particles smaller than 35 nm, the spherical cap model leads to a lower elastic penalty for the coherent existence of an interface between the α and β phases. It is therefore reasonable to assume that

cubic nanoparticles larger than 35 nm will still maintain coherency during hydrogen loading and unloading.”

Supplementary material corresponding author has a email address typo.

This typo has been corrected.

Fig. S1 shows that there are rods in the synthesis batch. Did the authors look at the phase morphology in these rods?

The rods are pentagonally twinned particles. Along their length, they expose their {100} facets and are capped with {111} facets at the ends. We are currently drafting a publication regarding their hydrogen absorption properties; unlike the single-crystalline cubes, they appear to show phase coexistence in equilibrium. Studies of their dynamics are underway.

Small changes that I would like to see implemented:

- In the caption of Fig. 1, please discuss the different panels (a, b, etc). Are all scalebars the same?

The caption has been updated to reflect that all scalebars are the same. Also, a much more significant discussion of Figure 1 has been included. It is reproduced below:

The caption has been updated:

“Still frames of the phase transformation as followed with STEM for three different particles of sizes (a) 20 nm, (b) 36 nm, and (c) 43 nm accompanied by their respective HRTEM images. The correspondence between each region and its respective phase was verified using EELS. The dotted lines represent the approximate locations of the phase boundaries. In some images, the cube drifts out of the field of view briefly, thus resulting in the image being cut off. The scale bar is the same for all images. The particles shown in (a) and (b) show the presence of a single nucleus of the β phase whereas the particle in (c) shows the existence of two nuclei.”

- In the introduction line 39-41 a comparison between Pd-H and LFP and LTO is made. A more comparable battery electrode is actually LiNiMnO₄, which has both solid solution and two-phase behavior (like Pd-H). LFP and LTO are typically thought of as exclusively two phase systems.

The sentence has been updated to exclude LFP and LTO and instead include LNMO. We have also included the appropriate references.

We have replaced:

“... with physics that closely parallel those of Li intercalation and deintercalation in LiFePO₄ and LiTiO₂”

with

“... with physics that closely parallel those of Li intercalation and deintercalation compounds, such as LiNiMnO₄.”

Clarification questions:

- for the Fig. 2 results. I assume the electron beam is rastered across the particle and the diffraction taken at each point in order to make the images in Fig. 2b-g?

The images are acquired using dark field microscopy. In this technique, an objective aperture is introduced into the diffraction plane to selectively allow through the diffracted intensity at a particular spot. The intensity from this spot is then used to create the image. In our case particularly, we use the objective aperture to select the intensity diffracted to a particular reciprocal lattice point by either the α or β phase. The following passage has been added to the supplementary information section about dark field microscopy:

In this technique, an objective aperture is introduced in the diffraction plane to select intensity that arises from a diffraction spot. The image is then reconstructed using the electrons diffracting to the selected spot.

The passage is referenced in the text.

Reviewer #2 (Remarks to the Author):

The manuscript 'Direct visualization of hydrogen absorption dynamics in individual nanoparticles' the authors report on a fascinating experimental in-situ work using ETEM and EELS. The authors study the hydride formation of individual single-crystalline Pd-nanocubes ranging from 15 nm to 80 nm in edge length. According to these results the hydride first forms in the cube corners and then the front propagates mainly parallel to the surface planes, e.g. in $\langle 100 \rangle$ -direction. This can be seen for all nanocube sizes, even for the smallest ones. The interface between the two phases inside the cube contains defects. This is supported by the finding of two well-separated hydride peaks in the selected diffraction pattern which are rotated by about 1.5° with respect to the other. The defects are gone after phase transformation. A similar behavior was found for all nanocube sizes. This is the first time of direct visualization of hydride fronts inside individual nanocubes of sizes down to 15 nm. While hydride nucleation in corners and two-phase coexistence in Pd nanocubes of about 80 nm side length and larger was reported via CXDI method by Ulvestad et al. recently, the here shown ETEM results step strongly forward. They are on much smaller nanocubes sizes and get new insights. They hint on defects accompanying the propagating front, surprisingly even for the small nanocubes. These defects affect the energy required to drive the phase transformation and are, thus, important for application. The results verify that the sub-surface layer which covers the whole nanocube (maybe not the bottom) was discussed to stimulate the hydride growth, has minor impact on the transformation behavior, as a planar front propagates in the nanocube. The results are very exciting and also, as I assume, difficult to achieve from the experimental part. The formation process is interesting and of general interest, not only for hydride formation in Pd-nanocubes – but also for other nanomaterials that sorb atoms on interstitial sites or by intercalation. For all these reasons I strongly support publication of this manuscript in Nature Materials.

Minor comments:

Abstract: The present version of the abstract highlights too much on the aspect that defects 'self

heal' after transformation – which is from my perspective not surprising, as the defects escort the front in the region of heaviest deformation. Once the front has passed the nanocube, the defects (I assume the authors mean a dislocation?) should also have left the sample. On the other hand it is known that defects are not stable in nanosized samples and have extremely fast kinetics. It is even difficult to visualize them in tensile test experiments on free samples – just reacted dislocations can be seen inside the samples (see for ex. B. Roos, B. Kapelle, G. Richter and C. A. Volkert, *Appl. Phys. Lett.* 105, 201908 (2014)).

We agree with the reviewer that this aspect was too prominent in the abstract. Following also the comments of Reviewer 1, we have updated the discussion of the observed reduced crystallinity during hydrogen uptake. In fact, while we consistently observe a 1.5° tilt between the α and β phases in the frozen reaction intermediates, the electron diffraction data don't provide strong supporting evidence for the presence of a dislocation. Such a small tilt could also be due to strain originating from a coherent interface between α and β . We have updated our discussion in the main text to reflect both these possibilities. The updates are reflected in the response to reviewer 1.

We had previously included the claim to highlight the ability of nanomaterials to remove defects that form inside them. The included reference is an interesting account of defect formation and annihilation in nanoscale systems, which are typically thought to be too difficult to form in nanoscale systems under reasonable stresses. The mechanisms discussed in the reference could help explain the source of dislocations in our samples.

Lines 69-82: This part informs about the results in advance. It is too lengthy and the reader is expecting experimental data first. This part further leads to information duplication. The authors should avoid this part.

This section has been shortened. The resulting passage is reproduced below:

“Here, we first use a combination of STEM and EELS to image the hydrogen absorption process in single crystalline Pd cubes in real time. We find that the reaction proceeds through a nucleation-and-growth pathway where the β phase nucleates in one or more corners of the cube before establishing a (100) phase front. We then examine nanocubes using DF imaging and SAED after freezing the reaction while it is in progress to examine the various reaction intermediates in greater detail. This analysis suggests the development of a lattice misorientation, which disappears upon completion of the transformation. SAED patterns of representative particles that have been loaded and unloaded twice show that the diffraction spots sharpen upon loading, further underscoring that the solute absorption process can reverse the crystal quality degradation induced by the α to β transformation.”

The impact of strain energy on the phase transformation behavior was recently modelled by Wagner et al. *Int. J. Hydr. Energy* 41, 2727(2016), on Pd-H thin films. The contribution of the strain energy to the isotherm shifts the chemical potential and affects the hydride formation tremendously – this explains why nanocube corners are hydride nucleation centers and why it is important to keep the system mainly stress-free for storage applications.

We agree that strain plays a significant role in the thermodynamics of PdH_x . We studied some of

the impacts of strain such as altered absorption pressure in our prior work (Baldi, Nature Materials 2014; Narayan Nature Materials 2014). The localization of hydrogen into the corners is likely to be favorable due to the tensile strain previously measured there. We note that favorable absorption through 111-like facets could also result in corner nucleation.

Reviewer #3 (Remarks to the Author):

The manuscript describes a detailed in-situ and ex-situ TEM/STEM characterization of the hydrogenation of palladium nanoparticles. The work is a fantastic illustration of what can be accomplished by in-situ TEM if it is performed under controlled illumination conditions. The results are extremely well presented, with the summary in the main text and the details of how the experiments were performed in the supplementary (every question I had about the experiments in the main text I found answered in the supplementary). The paper outline the experimental conditions and what assumptions go into the analysis in full detail that will allow anyone to reproduce the results. The analysis and the conclusions are all that one could want and expect.

Although I feel the paper should definitely be published, there is one area where I thought the authors could include a better discussion in the manuscript:

The paper starts with a description of the need to go to nanoparticles because of the ability to remove defects. However, in the results of figure 4 and supplementary data, there doesn't seem to be much of an effect on cycling in the particle size ranges the authors studied - they all appear to be small enough. What would be the maximum size where the authors feel there would be an effect? Would they be able to see a rate change or structure change specifically with their in-situ/ex-situ methods or would there have to be interpretation based on other results (in the manuscript, mechanisms proposed from other methods are used to corroborate the observations).

Although we do not have a quantitative estimate as to the size corresponding to the onset of bulk-like properties, we estimate that it is likely to be larger than 100 nm. A size of 100 nm appears to still be highly crystalline, as evidenced by the work of Ulvestad et al. (Nature Communications 2015). In that work, the authors hydride and then dehydride a 100 nm palladium particle and still are able to get high quality x-ray diffraction data to reconstruct the strain distribution inside a nanoparticle.

A possible transition to an incoherent mechanism could allow for differently oriented phase boundaries. The effects of persistent dislocations inside the structure will likely manifest themselves most clearly in a consistent widening of the diffraction spots with cycling. The coherency is likely maintained due to the lower stress present in the spherical cap model as compared to the spherical shell. We have now updated the text to better highlight how our results support a spherical cap model, which leads to a lower elastic penalty for coherent interfaces between α and β phases. Our on-going work is aimed at helping to address the reviewer's questions regarding transformation kinetics with even larger particles.

REVIEWERS' COMMENTS:

Reviewer #1 (Remarks to the Author):

I am happy with all of the changes and enthusiastically recommend publication.

Reviewer #2 provided comments to the Editor only, in which s/he recommends publication.

Reviewer #3 (Remarks to the Author):

The authors have made modifications to the manuscript that, as far as I can tell, have addressed the criticisms of the referees. I found it refreshing that the authors did not completely bow to the comments of the referees in order to be published and maintained the discussion in terms of what could be reliably determined from the data and what was open to discussion. I continue to believe that this is a fantastic demonstration of what can be accomplished by environmental TEM, provides unique and important results, and therefore the manuscript should be published.